# The p53-Driven Anticancer Effect of *Ribes fasciculatum* Extract on AGS Gastric Cancer Cells

**DOI:** 10.3390/life12020303

**Published:** 2022-02-17

**Authors:** Myeong-Jin Kim, Hye-Won Kawk, Sang-Hyeon Kim, Hyo-Jae Lee, Ji-Won Seo, Chang-Yeol Lee, Young-Min Kim

**Affiliations:** Department of Biological Science and Biotechnology, College of Life Science and Nano Technology, Hannam University, Deajoen 34054, Korea; jiny2415@naver.com (M.-J.K.); gyp03416@daum.net (H.-W.K.); ssho26@naver.com (S.-H.K.); canopus310@naver.com (H.-J.L.); ddongji28@naver.com (J.-W.S.); jefferysun4@naver.com (C.-Y.L.)

**Keywords:** *Ribes fasciculatum*, AGS, metastasis, anoikis resistance, apoptosis, EGFR, p53

## Abstract

Cancer metastasis is directly related to the survival rate of cancer patients. Although cancer metastasis proceeds by the movement of cancer cells, it is fundamentally caused by its resistance to anoikis, a mechanism of apoptosis caused by the loss of adhesion of cancer cells. Therefore, it was found that inhibiting cancer migration and reducing anoikis resistance are important for cancer suppression, and natural compounds can effectively control it. Among them, *Ribes fasciculatum*, which has been used as a medicinal plant, was confirmed to have anticancer potential, and experiments were conducted to prove various anticancer effects by extracting *Ribes fasciculatum* (RFE). Through various experiments, it was observed that RFE induces apoptosis of AGS gastric cancer cells, arrests the cell cycle, induces oxidative stress, and reduces mobility. It was also demonstrated that anoikis resistance was attenuated through the downregulation of proteins, such as epidermal growth factor receptor (EGFR). Moreover, the anticancer effect of RFE depends upon the increase in p53 expression, suggesting that RFE is suitable for the development of p53-targeted anticancer materials. Moreover, through xenotransplantation, it was found that the anticancer effect of RFE confirmed in vitro was continued in vivo.

## 1. Introduction

Cancer is the leading cause of death in modern society [1]. Cancer cells grow indefinitely, destroying normal organ functions and causing bleeding, infection, and thrombosis, eventually leading to death [2,3]. As a treatment method for cancer, the cancer is removed through surgery, and the growth of cancer cells is suppressed or killed using radiation therapy or anticancer drugs [4,5,6]. However, treatment of cancer is difficult, and metastasis is a major problem, among others. Cancer metastasis is most closely related to the survival rate of cancer patients, which is known to be difficult to treat. Therefore, there is a need for continuous research on the inhibition of cancer metastasis [7,8,9].

Epidermal growth factor receptor (EGFR) is known to be overexpressed in cancer cells, which recruits surrounding membrane proteins to lipid rafts, such as Caveolin-1, and phosphorylates them for signal transduction [10,11,12]. Signal transduction by EGFR is involved in the proliferation, survival, and metastasis of cancer cells. EGFR activates Src, a non-receptor protein tyrosine kinase, and Src is known to regulate proteins directly involved in cell proliferation and survival, such as Akt [13]. Akt is involved in the activity of mTOR to regulate cell proliferation and interacts with MDM2 to suppress the expression of cancer suppressors, such as p53, resulting in survival [14,15]. Therefore, overexpression of EGFR in cancer cells inhibits apoptosis and induces proliferation, which is known as a major means of enhancing anoikis resistance.

Although cancer cell metastasis is initiated by cancer cell migration and invasion, it is accompanied by increased angiogenesis and permeability by VEGF and acquisition of immune evasion against T-cells of disseminated tumor cells [16,17,18,19,20,21,22]. However, the fundamental problem of cancer metastasis is that cancer cells are resistant to anoikis. In general, when cells lose adhesion to the extracellular matrix (ECM), the activation of integrin is reduced, which leads to a decrease in the activities of FAK and Src. As a result, cell survival signaling pathways, such as PI3K/Akt, are reduced, which leads to apoptosis called anoikis [23,24]. However, since cancer cells are resistant to anoikis, they can survive and metastasize to other organs through blood vessels despite the loss of adhesion to the ECM. Cancer cells replace reduced integrin activity with overexpression of EGFR to enhance survival signals, such as Src and Ekr, thereby inhibiting anoikis [25,26,27]. Therefore, treatment strategies for cancer cell proliferation, apoptosis, and metastasis should include downregulation of EGFR and related proteins, such as Src and Caveolin-1.

Regarding the downregulation strategy of proteins involved in cancer metastasis, p53 is a notable cancer suppressor. p53 is a protein that inhibits carcinogenesis in normal cells. When cells are damaged, they repair and normalize, but normalization is difficult, and apoptosis is activated in cancerous cells [28,29]. However, p53 activates apoptosis in cells that are difficult to normalize and are cancerous. p53 inhibits anti-apoptotic proteins, such as the Bcl-2 family, and their inhibition activates pro-apoptotic proteins, such as Bak and Bax, to increase the mitochondrial membrane’s permeability. The increase in the mitochondrial membrane’s permeability activates caspase, leading to apoptosis. In addition, p53 is known to be an attractive cancer suppressor because it is known that it inhibits the formation of podosomes in cancer cells and contributes to the inhibition of metastasis due to reduced migration of cancer cells [30]. However, in most cancer cells, p53 activity is suppressed. This is because, depending on the activation of the Akt-MDM2 signaling pathway in cancer cells, p53 is ubiquitinated and subsequently degraded or rendered functionally defective by mutation [31]. Therefore, activating p53 in cancer cells can be suggested as one of the cancer treatment strategies.

In relation to p53 activity, natural bioactive substances are known to activate p53 in cancer cells [32,33]. In addition, in anticancer drug research, natural anticancer drugs are known to be effective with few side effects, so they are attracting attention in the development of anticancer drugs [34]. Therefore, while research on anticancer materials derived from natural products needs to be continued, *Ribes fasciculatum* is a noteworthy material.

*Ribes fasciculatum* is a plant that grows in East Asia and has been used as a medicinal plant to treat allergies in Korea. According to recent studies, *Ribes fasciculatum* is known to have anti-obesity, antioxidant, anti-aging, and allergy-relief effects; further, the anti-allergic effect is known to be closely related to cancer cell apoptosis [35,36,37,38,39,40,41,42]. Therefore, we suggest that *Ribes fasciculatum* has high anticancer bioactivity potential.

In this study, the mechanism of inducing apoptosis and inhibiting the metastasis of cancer cells by *Ribes fasciculatum* was investigated, and the association of p53 in anticancer efficacy was confirmed in vitro. In addition, it was also examined whether the anticancer activity of *Ribes fasciculatum* continued in vivo through xenotransplantation in Balb/c-nu/nu mice.

## 2. Materials and Methods

### 2.1. Reagents

Aerial parts of *Ribes fasciculatum* grown in Chungju (Korea) were ground and used. In total, 100 g of *Ribes fasciculatum* powder and 800 mL of 94.5% EtOH (Samchun Chemicals, Seoul, Korea) were mixed and suspended at RT for 72 h; then, EtOH was evaporated to obtain *Ribes fasciculatum* extract (RFE), which was stored at −20 °C. RFE was dissolved in DMSO (Samchun Chemicals, Seoul, Korea) according to the concentration to be used. Thiazolyl blue tetrazolium bromide (MTT powder, cat. 298-93-1) and pifithrin-α (PFT, cat. p4359) were obtained from Sigma-Aldrich (Merck KGaA, St. Louis, MO, USA). Annexin V and Dead cell Reagent (MCH100105), Cell Cycle Reagent (Cat. MCH100106), Oxidative Stress Reagent (MCH100111), and MitoPotential solution (Cat. MCH100110) used for flow cytometry were obtained from Luminex (Austin, TX, USA), while the Matrigel matrix was obtained from (Cat. 356237, Corning, New York, NY, USA). The primary antibodies EGFR (2232), p-EGFR (2234), p-Src (2101), Caveolin-1 (3267), p-Caveolin-1 (3251), Akt (9272), p-Akt (4060), p-MDM2 (3521), p53 (9282), Bcl-2 (3498), Bak (6947), β-actin (4967), PARP (9542), and CyclinE1 (4129) and secondary antibodies Anti-Rabbit (7074) and Anti-Mouse (7076) were obtained from Cell Signaling Technology (Beverly, MA, USA). The primary antibodies Src (sc-8056), Bax (sc-7480), Caspase3 (sc-7272), and VEGF (sc-7269) were obtained from Santa Cruz Biotechnology (Dallas, TX, USA). The primary antibodies p-CDK2 (ab194868) and cleaved caspase3 (ab2302) were obtained from Abcam (Cambridge, UK).

### 2.2. Cell Culture

AGS (21739), A549 (10185), HCT116 (10247), and HepG2 (88065) were obtained from Korean Cell Line Bank (Seoul, Korea), while BxPC3 (CRL-1687) and Hs738 (CRL-7869) were obtained from American Type Culture Collection (ATCC; Manassas, VA, USA). AGS, A549, HCT116, and BxPc3 were supplied with Roswell Park Memorial Institute-1640 (RPMI-1640, Cat. LM11211301; Welgene, Gyeongsan-si, Korea), containing 10% FBS and 1% penicillin streptomycin; HepG2 and Hs 738 were supplied with Dulbecco’s modified Eagle’s medium (DMEM, Cat. SH30243.01, HyClone Laboratories Inc., Marlborough, UK), containing 10% FBS and 1% penicillin streptomycin. The cells were incubated at 37 °C in 5% CO_2_ and sub-cultured using trypsin-EDTA every 2 days.

### 2.3. MTT Assay

In total, 1 × 10^5^ cells/mL of cells were seeded in a 12-well culture plate, incubated for 24 h at 5% CO_2_, 37 °C, and then treated with RFE. After 12 h, 24 h, and 48 h of RFE treatment, 40 µL of MTT solution (5 mg/mL) was added and incubated for 60 min. Then, the culture medium was removed, washed once with PBS, and 150 µL of DMSO was added to dissolve the formazan salt. A volume of 100 µL of DMSO dissolved in formazan salt was transferred to a 96-well plate, and then, the absorbance was measured at a wavelength of 595 nm to confirm cell viability.

### 2.4. Annexin V Staining

After seeding the AGS cells in a 6-well culture plate (5 × 10^5^ cells/mL), we incubated them for 24 h. Moreover, we treated them with RFE and incubated them for 24 h. Then, the cells suspended in trypsin-EDTA were collected and washed once with PBS. Afterwards, 100 µL of Annexin V and Dead cell Reagent was added to 100 µL of cells suspended in low serum media and incubated at 37 °C for 20 min. Thereafter, the cell apoptosis state was measured using a Muse cell analyzer (Luminex, Austin, TX, USA).

### 2.5. Cell Cycle Arrest Confirmation

Thereafter, the cells were suspended with trypsin-EDTA, washed with PBS, and then, 500 µL of cold 70% EtOH was added and left to react overnight. After removing EtOH and washing with PBS, 200 µL of Cell Cycle Reagent was added to the cells suspended in PBS and was left to react at room temperature (RT) for 30 min. Thereafter, the cell cycle state of the cells was observed using a Muse cell analyzer (Luminex, Austin, TX, USA).

### 2.6. Cell Oxidative Stress Measurement

AGS cells were seeded in a 6-well culture plate (5 × 10^5^ cells/mL) and cultured for 24 h. Moreover, after treatment with RFE, they were incubated for 24 h. The cells were then suspended with trypsin-EDTA and washed with PBS. A volume of 190 µL of oxidative stress working solution was added to 10 µL of cells suspended in PBS and incubated at 37 °C. After that, the oxidative stress of the cells was measured using a Muse cell analyzer (Luminex, Austin, TX, USA).

### 2.7. Measurement of Mitochondrial Membrane Potential

AGS cells were seeded in a 6-well culture plate (5 × 10^5^ cells/mL) and cultured for 24 h. Moreover, they were treated with RFE and incubated for 24 h. Thereafter, the cells were suspended in trypsin-EDTA, collected, and washed with PBS. A volume of 100 µL of MitoPotential working solution was added to 100 µL of cells suspended in PBS and then incubated at 37 °C for 20 min. Then, 7-AAD was added and incubated at RT for 5 min. Moreover, we measured the mitochondrial membrane potential of cells using a Muse cell analyzer (Luminex, Austin, TX, USA).

### 2.8. Transwell Assays

We put the Matrigel diluted in cold RPMI-1640 into a 24-well hanging chamber (Cat. 35224; SPL, Pocheon-si, Korea) and left it to react at 37 °C for 2 h. After removing the medium from the chamber and washing with PBS, we suspended the cells in a serum-free medium (2 × 10^5^ cells/mL were put in the chamber). We added a serum-containing medium to the bottom of the 24-well plate and incubated everything for 24 h. After removing the media from the bottom, we fixed the cells with 10% formalin at 4 °C for 1 h. After that, we removed all the solutions from the plate, stained the cells using 0.1% crystal violet, and washed the chamber 3 times with PBS. We then removed the cells from the upper part of the chamber with a clean cotton swab. Then, crystal violet was dissolved with 30% acetic acid, and the invaded cells were measured by measuring absorbance at the 590-nm wavelength. The migration assay proceeded without forming a matrix on the chamber.

### 2.9. Western Blotting

AGS cells were seeded (5 × 10^5^ cells/mL) in a 6-well plate and incubated for 24 h; then, they were treated with RFE and incubated for 24 h. Thereafter, the cells were lysed using RIPA lysis buffer (ForBioKorea, Korea) containing a phosphatase inhibitor cocktail and then centrifuged at 14,000 rpm, at 4 °C, for 20 min to separate cell proteins. The extracted proteins were quantified through a Bradford assay, and 20 μg was electrophoresed on an 8–12% acrylamide gel. The separated proteins were transferred to a nitrocellulose membrane. Then, the membrane and primary antibodies were left to react overnight. After that, the membrane was left to react with the secondary antibody for 2 h; then, it was washed sufficiently, and the protein expression level was obtained using a chemiluminescent substrate (cat. 34579, Thermo Fisher, Middlesex, MA, USA). Western blot results were semi-quantified using ImageJ (National Institutes of Health, Bethesda, MD, USA).

### 2.10. Xenograft Model

Male, 4-week-old Balb/c nu/nu mice were obtained from Envigo (Cumberland, VA, USA) and bred in a specific pathogen-free (SPF) barrier system. Experimental animals were bred in an environment with a temperature of 21 ~ 25 °C, 45 ~ 55% humidity, and a 12-h light cycle. Experimental animals were acclimatized to the environment for one week, and mice without weight change were selected and divided into 3 groups of 5 specimens each. In total, 2 × 10^6^ cells/0.25 mL of PBS-suspended AGS gastric carcinoma cells were injected subcutaneously in the axillary region between the right shoulder blade and chest wall. After cancer cell transplantation, from the time when the tumor size of each group reached about 50 mm^3^, RFE was administered at the same time every day, and the weight and tumor size of the experimental animals were measured every 2 days for 20 days. After 20 days of RFE administration, the experimental animals were sacrificed, and cancer tissues were extracted and fixed in 10% formalin. RFE was dissolved in PBS and administered intraperitoneally, while only PBS was administered to the vehicle group. All the animal experiments were conducted with the approval of the Hannam University Animal Experimental Ethics Committee (2021-13(HNU2021-13); Daejeon, Korea).

### 2.11. Tunnel Assay

Cancer tissues fixed in 10% formalin were sequentially dehydrated using EtOH (70%, 80%, 90% and 99%) and Xylene, then fixed using paraffin. Then, the paraffin block-fixed tissue was cut to a thickness of 4 μm. After the paraffin was removed, the tissues were treated with the ApopTag Peroxidase in situ Apoptosis Detection Kit (Vector Laboratories, San Francisco, CA, USA). We added DAB solution and left the samples to react for 7 min, and then, we left them to react with hematoxylin for 3 min. Intra-tissue apoptosis was quantitatively evaluated through ImageJ (National Institutes of Health, Bethesda, MD, USA) after imaging under a microscope.

### 2.12. Immunohistochemistry

The cancer tissue fixed with formalin and paraffin blocks was cut to a thickness of 4 μm and heated in an oven at 60 °C for 40 min. The paraffin was removed sequentially with Xylene and EtOH (99%, 90%, 80%, and 70%). Then, the tissues were treated with 3% H_2_O_2_ for 15 min, and non-specific binding was prevented with 5% BSA. Then, the sections were incubated with primary antibodies at 4 °C for 60 min. After leaving them to react with the secondary antibody for 30 min, we added a DAB solution for 7 min to let color develop. Then, the cancer tissue sections were stained with hematoxylin. The protein expression in tissues was quantitatively evaluated through ImageJ (National Institutes of Health, Bethesda, MD, USA) after imaging under a microscope.

### 2.13. Statistical Analysis

All the experiments were repeated at least 3 times. The mean of the experimental results, the calculation of standard error and significance evaluation using a *t*-test and ANOVA (analysis of variance) were performed using the SPSS 21.0 program (IBM-SPSS, Chicago, IL, USA). The significance among groups was divided into levels of *p* < 0.05, *p* < 0.01, and *p* < 0.001.

## 3. Results

### 3.1. Confirmation of Cytotoxicity of RFE

To determine the cytotoxicity of *Ribes fasciculatum* extract (RFE) to cancer cells, various cancer cell lines, such as AGS (gastric cancer cells), A549 (lung cancer cells), HCT116 (colon cancer cells), BxPC3 (pancreatic cancer), and HepG2 (liver cancer cells), were treated with RFE at 60 µg/mL. Moreover, cell viability was assessed by the MTT assay. After 24 h of RFE treatment, a significant decrease in cell viability was observed in AGS, A549, and BxPC3, and a significant decrease in cell viability was confirmed in all cancer cell lines after 48 h of treatment. Among them, the survival rate of AGS gastric cancer cells decreased to 48% at 24 h of RFE treatment and decreased to 18% at 48 h treatment, confirming that the cell viability of AGS was reduced the most by RFE (Figure 1A,B). When AGS cells were treated with RFE at various concentrations, it was demonstrated that the cell viability decreased in a dose-dependent manner, and it was observed that the cell viability was reduced by 36% at a concentration of 90 µg/mL (Figure 1C). However, when RFE was treated in normal gastric cells, cytotoxicity was not observed at any concentration (Figure D). Through this, it was found that RFE has cancer cell-specific cytotoxicity, and particularly, it shows strong toxicity to AGS.

### 3.2. Effects of RFE on Inducing Apoptosis and Cell Cycle Arrest

In order to confirm the anticancer effect of RFE on AGS cells, various flow cytometry analyses were performed. First, it was confirmed by MTT assay that RFE exhibits strong cytotoxicity to AGS cells. To confirm that this cytotoxicity is due to apoptosis, RFE was treated on AGS cells for 24 h. Moreover, Annexin V staining was performed. Annexin V stains phosphatidylserine that is exposed on the outside of the cell membrane when apoptosis occurs in cells, and through this, apoptosis is measured. As a result of the experiment, it was found that apoptosis was increased in an RFE concentration-dependent manner (Figure 2A). In addition, the mitochondrial membrane potential decreased in an RFE concentration-dependent manner, confirming that RFE-induced apoptosis was caused by the loss of the mitochondrial membrane potential (Figure 2B). In addition, an increase in oxidative stress was observed by RFE. Excessive oxidative stress in cancer cells is known to cause cancer cell death, so it can be said that the increase in oxidative stress by RFE is an effect that contributes to toxicity to AGS cells (Figure 2C) [43]. In addition, it was observed that the G0/G1 phase of cells increased in an RFE dose-dependent manner, confirming that RFE arrests the cell cycle of AGS cells. Therefore, it was confirmed that RFE induces apoptosis and oxidative stress in AGS gastric cancer cells and reduces proliferation by arresting the cell cycle (Figure 2D).

### 3.3. The Effect of Reducing Migration and Invasion of RFE

A transwell assay was performed to confirm the effect of RFE on cell mobility, which is directly related to cancer metastasis. As a result of the experiment, it was confirmed that the migration of AGS cells decreased in a dose-dependent manner when RFE was treated (Figure 3). Among them, it was observed that the migration of cells that invaded the Matrigel matrix was more inhibited when RFE was treated. Simple migration of AGS cells was reduced by 22% compared to control at a concentration of 30 µg/mL, and invasion was reduced by 36%. Through this, it seems that the decrease in cell migration by RFE can inhibit invasion more strongly.

### 3.4. Effect of Protein Expression Change by RFE

In this study, AGS cells were treated with RFE at concentrations of 30 and 60 µg/mL and left to react for 24 h to confirm the change in protein expression in AGS cells through Western blotting. EGFR activity is a major contributor to anoikis resistance. As a result of the experiment, the expression of EGFR was almost constant, but the expression of the activated form, p-EFGR, was greatly reduced according to the RFE concentration, and the expression level of p-Src was also greatly reduced (Figure 4A,B). In addition, the activity of Caveolin-1, which helps signal transduction according to the activity of EGFR, also decreased according to the concentration of RFE. With the decrease in p-Src, it was also observed that the activity of Akt, which is involved in cell survival and proliferation, decreased. Accordingly, a cell cycle arrest tendency was observed according to a decrease in CyclinE1 and an increase in p-CDK2, while the expression of VEGF, which induces angiogenesis, was also decreased [44,45,46]. In addition, it was observed that the expression level of p53 was significantly increased because the expression of p-MDM2 was decreased with the decrease in Akt activity. As the expression of p53 increased, a fragment of PARP that repairs DNA was observed, and the expression of Bax and Bak was increased, while the expression of Bcl-2 was decreased. As a result, it was revealed that the mitochondrial membrane permeability was increased, and Caspase3 was cleaved and activated to induce apoptosis. Through these protein expression changes, it was confirmed that RFE lowered the anoikis resistance of AGS cells, induced apoptosis, and inhibited cell growth.

### 3.5. Confirmation of p53 Dependence of Anticancer Effect by RFE

p53 is known as a cancer suppressor protein, and its expression level in AGS was significantly increased upon RFE treatment. In order to check how predominantly p53 activity occurs in the anticancer effect of RFE, the p53 inhibitor pifithrin-α (PFT) and RFE were concurrently treated with AGS cells to conduct an experiment. As a result of the experiment, when RFE and PFT were treated in parallel, a 31% increase in cell viability was observed compared to when RFE was treated alone, and it was shown that the mitochondrial membrane potential was significantly increased by inhibition of p53 (Figure 5A,B). In addition, changes were observed in protein expression through Western blotting. When p53 was inhibited by PFT, the activity of EGFR, Src, and Caveolin-1 was increased, and the expression of apoptosis-related proteins below p53 was also changed in the tendency of cells to survive (Figure 5C). Therefore, it was demonstrated that the anticancer effect by RFE was dependent on p53.

### 3.6. Confirmation of Anticancer Effect of RFE In Vivo through Xenotransplantation

In order to confirm whether the anticancer effect of RFE observed in vitro was also effective in vivo, xenotransplantation was performed using BALB/c nude mice. When the transplanted tumor volume of each group reached 50 mm^3^, RFE was administered intraperitoneally for 20 days. As a result of the experiment, no change in the weight of mice was observed according to the RFE treatment, and it was found that the tumor volume was decreased in an RFE dose-dependent manner of (Figure 6A,B). Compared to the vehicle group, the tumor volume of the RFE 60 mg/kg/day administration group was reduced by 27%, and it was revealed that the tumor volume decreased by 43% in the 90 mg/kg/day administration group. Through this, it was demonstrated that RFE inhibited tumor growth. The TUNEL assay was performed to confirm the induction of apoptosis in the tumors extracted by sacrificing the experimental animals. The TUNEL assay can confirm DNA fragmentation caused by apoptosis, and as a result of the experiment, it was confirmed that the area where apoptosis occurred in cancer tissue increased according to the concentration of RFE. In addition, immunohistochemistry (IHC) was performed to confirm changes in protein expression in tissues to observe the expression of p-EGFR and p53 (Figure 6C). The expression of p-EGFR in the tissue was observed to have decreased with the increase in RFE concentration, and as the expression of p53 was observed to have increased, it was demonstrated that the anticancer effect of RFE confirmed in vitro was continued in vivo.

## 4. Discussion

Research on drugs for the treatment of cancer has already greatly progressed, and many effective anticancer drugs are being used for treatment on the market. Nevertheless, the reason why research on the development of anticancer drugs is continuing is that the individual differences in response to anticancer drugs are large. It is necessary to diversify cancer treatment strategies due to differences in individual metabolic ability, cancer mutation, drug resistance, and side effects. In this regard, the development of various anticancer drugs is the basis for personalized cancer treatment [47,48,49,50]. As part of research aimed to expand the basis of cancer treatment, this study also presents a basic study for the development of medicinal products of *Ribes fasciculatum*.

Cancer cells inhibit programmed cell death and metastasize through migration to nearby organs or invasion into blood vessels. Therefore, controlling it helps to improve cancer treatment and the survival rate of patients [16,25,51]. We conducted an experiment to confirm whether *Ribes fasciculatum* is suitable for the development of anticancer drugs under the above conditions. *Ribes fasciculatum* is known to have several beneficial bioactive effects, but its anticancer effects are unknown. However, when several cancer cells were treated with *Ribes fasciculatum* extract (RFE), it was found that the viability was reduced, and among the cell lines tested, AGS gastric cancer cells, known as metastatic cells, showed significant toxicity. The cell viability of AGS by RFE decreased in an RFE-dose-dependent manner and was observed to be reduced to 36% at a concentration of 90 µg/mL, but RFE did not show cytotoxicity in normal gastric epithelial cells. Through this, it was demonstrated that RFE is not toxic to normal cells and has cancer cell-specific cytotoxicity. Moreover, through flow cytometry, it was confirmed that the cancer cell-specific cytotoxicity of RFE is due to apoptosis, which causes the loss of the mitochondrial membrane potential, thereby confirming the anticancer efficacy of RFE in inducing apoptosis. In addition, by RFE, cells in the S and G2/M phases during cell proliferation were decreased, and cells in the G0/G1 phase were increased. In addition, by confirming the decrease in CyclineE1 expression and the increase in p-CDK2, it was revealed that RFE arrests the cell cycle of AGS cells and inhibits proliferation.

Since metastasis of cancer cells requires mobility, a decrease in the mobility of cancer cells is an indicator of a decrease in metastasis [52]. In order to confirm the effect of RFE in relation to metastasis inhibition, a wound-healing assay as well as a migration assay and an invasion assay that can confirm the mobility of cancer cells were performed. Through the reduction in wound healing, the effect of RFE in terms of decreasing the proliferation and migration of AGS cancer cells was demonstrated; furthermore, through migration and invasion assays, it was observed that RFE inhibited invasion more than simple cell migration. In addition, VEGF is known to increase vascular permeability, indicating that the decrease in VEGF expression shown by Western blotting was also an effect contributing to the reduction in cancer metastasis. Therefore, it is suggested that RFE is a substance that effectively acts on the reduction in metastasis by invasion of cancer cells [19,20,21].

The identification of molecular signaling pathways in drug research is essential because it allows us to understand the mechanisms of drugs and predict their effects. In this study, changes in molecular signaling were observed through Western blotting, and it was confirmed that the activity of EFGR decreased when RFE was treated with AGS cells. As the activity of EGFR decreased, the activities of Src and Akt were decreased. Accordingly, the Akt-MDM2 pathway was down-regulated, and p53 expression was increased. The increase in p53 expression inhibited Bcl-2 and activated proteins, such as Bax and Bak, and it was observed that the apoptosis cascade proceeded by the reduction in the mitochondrial membrane potential. Moreover, since Caveolin-1 is known to contribute to EGFR and TGF beta-1-mediated anoikis resistance, it can be said that the decrease in cell survival signal by reduction of EGFR, Src, and Caveolin-1 activity reduces anoikis resistance [53,54]. This suggests that RFE can inhibit anoikis resistance in AGS cells and effectively induce apoptosis.

Here, we focused on the expression of p53, which is known as an important cancer suppressor. Regarding the cause of the increase in the expression of p53 by RFE, it was demonstrated that the expression of p53 was increased due to the decrease in the activity of MDM2. However, since excessive oxidative stress in cancer cells expresses p53, the increase in oxidative stress in AGS cells by RFE also seems to increase p53 expression [55,56]. Therefore, it is suggested that RFE is optimized to increase the expression of p53 in cancer cells, and it is considered that p53 is significantly related to the anticancer effect of RFE. Therefore, in order to confirm this, an experiment was conducted using pifithrin-α (PFT), an inhibitor of p53. As a result of the experiment, when p53 was inhibited by PFT, the viability of AGS cells increased, the expression of pro-apoptotic proteins was decreased, and the mitochondrial membrane potential was restored, confirming that RFE-induced apoptosis in AGS cells was p53-dependent. In addition, it was observed that the activity of proteins involved in anoikis resistance, such as EGFR, Src, and Cavolin-1, increased by inhibition of p53. Although p53 is not involved in direct inhibition of EGFR, it is known to increase sensitivity to EGFR-Tyrosin kinase inhibition. Therefore, the activity of p53 enhances the inhibition of EGFR activity by RFE, and thereby EGFR and Src activity are regulated [57,58,59]. Since increasing the sensitivity to inhibition of EGFR is one of the methods to reduce resistance to EGFR-targeted anticancer drugs, RFE has shown the potential to help overcome anticancer drug resistance. In addition, as the decrease in the activity of Caveolin-1 was observed by the decrease in the activity of p53, the possibility that p53 could contribute to the regulation of the activity of Caveolin-1 was revealed. This can suggest a direction for future research on the development of anticancer drugs using RFE. In addition, with regard to cell migration ability, since p53 reduces the migration ability of Src-mediated cancer cells, the expression of p53 by RFE appears to be involved in cell migration ability. Therefore, it was revealed that the various anticancer effects of RFE were concentrated on the activity of p53, suggesting that RFE is an effective material for a cancer treatment strategy targeting p53 (Figure 7). 

In addition, xenograft model experiments were conducted. By observing the growth inhibition of cancer tissue, induction of apoptosis, and changes in the expression of p-EGFR and p53, it was confirmed that the anticancer effect of RFE in vitro was continued in vivo. It is shown that RFE effectively inhibits cancer even in vivo.

## 5. Conclusions

In this study, the anticancer effect of RFE was demonstrated through several experiments. RFE is shown to reduce cancer cell mobility and anoikis resistance, thereby lowering metastasis and inducing apoptosis of cancer cells. In addition, this effect was found to have a concentrated effect on p53 expression by RFE, which suggests a new research direction for the study of anoikis-resistance treatment using natural products. The identification of the compounds contained in *Ribes fasciculatum* is still insufficient, and studies on the anticancer effects of the revealed compounds have not been conducted or are insufficient [36,37,60]. Therefore, this study suggests the possibility of searching for new compounds with anticancer effects in *Ribes fasciculatum* or research on the anticancer effects of known substances on metastasis and apoptosis, which could be the basis for developing anticancer drugs using *Ribes fasciculatum*.

## Figures and Tables

**Figure 1 life-12-00303-f001:** Confirmation of cytotoxicity of *Ribes fasciculatum* extract (RFE) by MTT assay. (**A**) Changes in cell viability by 12, 24, and 48 h when RFE was treated with various cancer cells. (**B**) Cell viability after 24 h of RFE treatment in various cancer cell lines. (**C**) Cell viability when AGS cells were treated with RFE at 10–90 ug/mL for 24 h. (**D**) Cell viability when Hs738 normal gastric epithelial cells were treated with RFE at 10–90 ug/mL for 24 h. The data were confirmed after repeating the experiment three times. Statistical analyses performed using a *t*-test. * *p* < 0.05, ** *p* < 0.01, and *** *p* < 0.001 compared with Group N.

**Figure 2 life-12-00303-f002:** Flow cytometry analysis of *Ribes fasciculatum* extract (RFE) 24 h after treatment with AGS cells. (**A**) Effects of inducing apoptosis by RFE using Annexin V and PI staining. (**B**) Changes in mitochondrial membrane potential by RFE. (**C**) Oxidative stress-inducing effect of RFE by flow cytometry analysis. (**D**) Changes in cell distribution of cell cycle states by RFE. The data were confirmed after repeating the experiment three times. Statistical analyses performed using a *t*-test. * *p* < 0.05, ** *p* < 0.01, and *** *p* < 0.001 compared with Group N.

**Figure 3 life-12-00303-f003:** Confirmation of reduction in mobility of AGS by *Ribes fasciculatum* extract (RFE). Effect of reducing migration and invasiveness of RFE on AGS cells through transwell assays. The data were confirmed after repeating the experiment three times. Statistical analyses performed using a *t*-test. * *p* < 0.05, ** *p* < 0.01, and *** *p* < 0.001 compared with Group N.

**Figure 4 life-12-00303-f004:** Changes in protein expression in AGS cells by *Ribes fasciculatum* extract (RFE). (**A**) Confirmation of protein expression by Western blotting: p-EGFR; EGFR; p-Src; Src; p-Caveolin-1; Caveolin-1; p-Akt; Akt; p-MDM2; p53; PARP; Bcl-2; Bax; Bak; Pro-Caspase3; Cleaved-Caspase3; CyclinE1; p-CDK2; VEGF; and *β*-actin. (**B**) Semi-quantitative analysis. The data were confirmed after repeating the experiment three times. Statistical analyses performed using a *t*-test. Figure 4B shows the concentration readings/intensity ratios of each band for all western blot readings. And the original western blot figure is attached as a Appendix A. * *p* < 0.05, ** *p* < 0.01, and *** *p* < 0.001 compared with Group N.

**Figure 5 life-12-00303-f005:** Changes in the anticancer effect of *Ribes fasciculatum* extract (RFE) on AGS cells according to p53 inhibition. (**A**) Confirmation of changes in cell viability by RFE during p53 inhibition through MTT assay. (**B**) Changes in mitochondrial membrane potential following p53 inhibition by flow cytometry analysis. (**C**) Changes in protein expression in AGS cells by RFE upon p53 inhibition: p-EGFR; p-Src; p-Caveolin-1; p53; Bcl-2; Bax; Cleaved Caspase3; and *β*-actin. The data were confirmed after repeating the experiment three times. Statistical analyses performed using a *t*-test. * *p* < 0.05, ** *p* < 0.01, and *** *p* <0.001 compared with Group N; ^#^ *p* < 0.05, ^##^ *p* < 0.01, and ^###^ *p* <0.001 compared with Group RFE.

**Figure 6 life-12-00303-f006:** Anticancer effect of *Ribes fasciculatum* extract (RFE) using xenograft model. (**A**) Changes in body weight for 20 days after administration of RFE to experimental mice. (**B**) Changes in volume of xenograft cancer tissue following administration of RFE. (**C**) Confirmation of apoptosis induction through TUNEL assay and changes in p-EGFR and p53 expression through immunohistochemistry (IHC) in xenograft cancer tissue treated with RFE for 20 days. All substances were administered intraperitoneally. Results were confirmed in five mice in each group. Statistical analyses performed using one-way and two-way analysis of variance (ANOVA). * *p* < 0.05, ** *p* < 0.01, and *** *p* < 0.001 compared with Group N.

**Figure 7 life-12-00303-f007:** The molecular regulatory mechanism underlying *Ribes fasciculatum* extract (RFE) in AGS cells. Continuous arrows indicate signal enhancement, dotted arrows indicate signal attenuation, and blocked arrows inhibit the activity of the corresponding protein. RFE inhibits the EGFR signaling pathway and increases the activity of p53.

## Data Availability

The data presented in this study are provided in this manuscript.

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
