# Peer review of "The p53-Driven Anticancer Effect of Ribes fasciculatum Extract on AGS Gastric Cancer Cells"

_life, 2022, doi:10.3390/life12020303_

Round 1

Reviewer 1 Report

An article titled: "The p53-driven anticancer effect of Ribes fasciculatum extract on AGS gastric cancer cells" presents through various experiments anticancer properties of Ribes fasciculatum extract. It is well designed and comprehensive research, including even in vivo presentation of anti-cancer properties of the extract.

My only concern is how many times were experiments repeated, since there is no information in the figure description

Author Response

Reviewer's Comment : My only concern is how many times were experiments repeated, since there is no information in the figure description

>>Thank you for your valuable comments to improve the manuscript. As you commented, the information in the figure did not explain how many times the experiment was repeated. This was mentioned in Section 2.14, but I thought it would be reasonable to explain it in the caption of the figure as you suggested, so I revised the manuscript to reflect this. The number of repetitions of the experiment was mentioned in each figure, and the number of animals per group of each experimental animal was mentioned in the in-vivo experiment. Thank you for reviewing the manuscript and for your kind comments.

Reviewer 2 Report

I’d like to share a few review comments with you as follows:

Major points

In Figure 3, shows the results of wound healing assay, migration assay and invasion assay by REF treatment in AGS gastric cancer cells.

However, it is to be not shown an exact-results.

Because, the results of Figure 3 show that the toxic effect of REF treatment on AGS gastric cancer cells, not the reduction of metastasis by REF treatment. (Especially, the results of REF 60µg/ml concentration treatment)

If authors want to confirm exact-results of metastasis on AGS gastric cancer cells treated with exact REF treatment using wound healing assay, migration assay or invasion assay, set a concentration that is not toxic to cells when setting the REF treatment concentration, and then the experiment should to do.

Minor points

In supple data, authors showed the uncropped western-blot band of pro-caspase and cleaved-caspase form bands separately. I would think that authors need to be show that both pro-caspase and cleaved-caspase form bands in whole membrane.

Author Response

Reviewer's Comment 1:

Major points

In Figure 3, shows the results of wound healing assay, migration assay and invasion assay by REF treatment in AGS gastric cancer cells.

However, it is to be not shown an exact-results.

Because, the results of Figure 3 show that the toxic effect of REF treatment on AGS gastric cancer cells, not the reduction of metastasis by REF treatment. (Especially, the results of REF 60µg/ml concentration treatment)

If authors want to confirm exact-results of metastasis on AGS gastric cancer cells treated with exact REF treatment using wound healing assay, migration assay or invasion assay, set a concentration that is not toxic to cells when setting the REF treatment concentration, and then the experiment should to do.

>> First, based on the following papers, I set the concentration of RFE in the experiment to confirm cytotoxicity and cell mobility.

Kang, Dong Young, et al. "Antitumor effects of ursolic acid through mediating the inhibition of STAT3/PD-L1 signaling in non-small cell lung cancer cells." Biomedicines 9.3 (2021): 297.

Kawk, Hye Won, et al. "Scaphium affine Ethanol Extract Induces Anoikis by Regulating the EGFR/Akt Pathway in HCT116 Colorectal Cancer Cells." Frontiers in Oncology 11 (2021).

Since these papers confirmed cell migration at a level of cytotoxicity similar to that of this manuscript, the corresponding dose was set in this experiment. As you suggested, I also agree that it is ideal to observe the mobility at a concentration that is not toxic to cells in order to confirm the complete decrease in cell mobility. However, signaling pathways such as Akt-mTOR are involved in cell migration in cancer cells, and RFE has been shown to inhibit Akt in this manuscript. In addition, the anticancer effect of RFE is concentrated on p53, which is known to inhibit the formation of podosoms, which are important for cancer cell migration, thereby reducing migration. Therefore, the complete isolation of RFE in terms of cytotoxicity and cell mobility is likely to be difficult. In addition, at a concentration of less than 10ug/ml, it is expected that the cell response by RFE will be small, and it seems that it will be impossible to observe cell migration at a concentration of less than 10ug/ml. However, even if the above is the case, I also agree that the concentration of 60ug/ml you mentioned is an excessive concentration setting to explain the mobility in the manuscript. So I would like to replace it with a result of concentrations of 20 and 40 ug/ml. But from your point of view, I would like to ask if my opinion is still highly controversial and is not convincing. If you can agree with my opinion to some extent, I would like to present the experimental results of concentrations of 20,40ug/ml. However, if you still think that it is still controversial when you look at it, we would like to conduct the experiment at a concentration of less than 10ug/ml, and if there are no results, we would like to exclude the section related to mobility from the manuscript.

You have already reviewed the manuscript and provided comments to improve it. Nevertheless, I am sorry, but I would like to ask for your opinion once again.

Reviewer's Comment 2:

Minor points

In supple data, authors showed the uncropped western-blot band of pro-caspase and cleaved-caspase form bands separately. I would think that authors need to be show that both pro-caspase and cleaved-caspase form bands in whole membrane.

>>I agree with your opinion. I provided the bands of pro-caspase3 and cleaved-caspase3 separately to suggest a clearer band. However, as you suggested, I agree that it is good to provide pro- and cleaved foams in one membrane. Therefore, the caspase3 western blotting of the manuscript was replaced with the data requested by you, and a new raw-western blot was attached.

Round 2

Reviewer 2 Report

You would like to replace it with a result of concentrations of 20 and 40 ug/ml.

► RFE concentration of 40ug/ml is an excessive concentration setting to explain the mobility.

I suggest that RFE concentrations setting of 10 and 30 ug/ml on migration.

In addition, adding to figure on the Schematic summary of a role of RFE (FIG. 7) in AGS gastric cancer cells.

It is unclear that results of RFE on the interaction of apoptosis and migration in AGS cells.

Author Response

Reviewer's Comment 1:

RFE concentration of 40ug/ml is an excessive concentration setting to explain the mobility. I suggest that RFE concentrations setting of 10 and 30 ug/ml on migration.

►Thanks for your kind suggestion. According to your opinion, the figure was replaced by using the experimental result data of 10 and 30ug/ml. However, in the case of wound healing assay, we had no choice but to remove it because it is currently impossible to experiment in our laboratory. Therefore, the manuscript was revised accordingly. Your suggestion has helped address the controversy over cytotoxicity and mobility

Reviewer's Comment 2:

In addition, adding to figure on the Schematic summary of a role of RFE (FIG. 7) in AGS gastric cancer cells. It is unclear that results of RFE on the interaction of apoptosis and migration in AGS cells.

►I agree with your opinion. My manuscript described the various anticancer effects of RFE. However, since the results were presented for too broad an anticancer effect, this tends to obscure the anticancer effect of RFE, which is mainly explained in the manuscript. So I was able to solve this weakness by adding a new figure as per your suggestion.

Round 3

Reviewer 2 Report

I confirmed that you describe new figure in your manuscript.